# Effects of Teaching Program Based on Teaching Games for Understanding Model on Volleyball Skills and Enjoyment in Secondary School Students

Maja Batez [1] , Tanja Petrušič [2], Špela Bogataj [3,4] and Nebojša Trajković [5,*]

1 Faculty of Sport and Physical Education, University of Novi Sad, 21101 Novi Sad, Serbia; majabatezns@yahoo.com
2 Faculty of Education, University of Ljubljana, 1000 Ljubljana, Slovenia; tanja.petrusic@pef.uni-lj.si
3 Department of Nephrology, University Medical Centre, 1000 Ljubljana, Slovenia; spela.bogataj@kclj.si
4 Faculty of Sport, University of Ljubljana, 1000 Ljubljana, Slovenia
5 Faculty of Sport and Physical Education, University of Niš, 18000 Niš, Serbia
* Correspondence: nele_trajce@yahoo.com

**Abstract:** This study investigated the effects of the Teaching Games for Understanding (TGfU) model implemented in physical education classes on volleyball skills and enjoyment in secondary school students. A total of 54 students (18 girls) from two classes participated in this study, of whom 28 (age = 15.5 ± 0.7 years) were randomized to a TGfU model (EXP) group and 26 (age = 15.7 ± 0.6 years) to a control group (CON) that maintained their usual physical-education activities. Four tests for volleyball skills were conducted: service, overhead, and forearm passing and setting. Additionally, the sport enjoyment questionnaire was used the first and the last week of intervention. Results from repeated measures analysis of variance (ANOVA) showed a significant interaction for overhead passing (F 1, 58 = 5.273, $p$ = 0.025, Partial $\eta^2$ = 0.083) and forearm passing (F 1, 58 = 4.641, $p$ = 0.035, Partial $\eta^2$ = 0.074). When examining the impact of TGfU program on service accuracy, there was a significant main effect for time ($p$ < 0.01) with both groups improving their result after the six-weeks intervention (EXP-ES = 0.32, % change = 9.1% vs. CON-ES = 0.57, % change = 14.4%). There were no significant time or group × time effects for setting ($p$ > 0.05). The EXP group showed significantly better results for enjoyment compared to the CON group ($p \leq$ 0.05). The findings show the effectiveness of the TGfU model of short duration (12 lessons) in an educational context to improve volleyball skills. We also highlight the importance of enjoyment during these classes compared to traditional physical education classes.

**Keywords:** teaching program; adolescents; technique; school

## 1. Introduction

Throughout history, teaching approaches in physical education (PE) were evolving and transforming [1]. New teaching approaches focus on using modified games, technical–tactical learning in similar sports, cognitive training, learning progression, teaching tactics before teaching technique, and problem-solving [2].

Looking from an educational perspective, teaching sports games is a significant part of the PE curriculum [3]. Furthermore, it was observed that the use of isolated techniques conquered teaching as a part of structured lessons [4]. With this approach, we isolate skills teaching and learning and later transfer them into the actual game [5,6]. Students reported bad experiences with PE teaching and learning processes and identified them as a barrier to participating in sports and other physical activities in their childhood and adolescence [7]. Furthermore, the teaching strategies have been recognized as the main limitation in supporting suitable development in the cognitive, psychomotor, and affective learning domains during PE lessons [8]. In the traditional teaching model, the gameplay is

only presented at the end of the lessons, and consequently, the isolated skill-drill students often perceive it as meaningless and boring [6].

In this regard, game-based pedagogy approaches have been promoted to improve physical fitness, skill execution, and decision-making in PE and sports teaching/coaching [1]. The Teaching Games for Understanding (TGfU) approach was presented as an alternative to the traditional content-orientated model [4], which includes skill and tactics learning throughout actual gameplay. TGfU generates an immense game understanding and increases motivation, physical activity levels, enjoyment, and engagement in PE lessons [9]. This model includes modified games to encourage decision-making in an active learning setting with strategic and tactical problems [10].

Previous studies examined the difference between the traditional technique learning model and the game-based model on performance in PE classes [11]. Students that were engaged in a game-based learning model had a significantly higher declarative knowledge than students involved in a technical teaching approach [3,12].

The problem of traditional learning methods is a lack of enjoyment and motivation. Therefore, this study aimed to determine if the TGfU approach is better than skill-oriented teaching to improve volleyball skills and increase enjoyment in secondary school students.

## 2. Materials and Methods

### 2.1. Participants

This was a cluster-randomized, interventional trial comparing TGfU school-based program with traditional physical education classes in adolescent students. Fifty-four adolescent students (18 girls) from two different classes in school from southern Serbia were included in the study, of whom 28 (age = 15.5 ± 0.7 years) were randomized to a TGFU school-based program (EXP) group and 26 (age = 15.7 ± 0.6 years) to a control group (CON) that maintained their usual physical education activities.

To be included in the study, participants had to be between 14 and 16 years old, be free of any medications that could affect the results, not have medical problems, and not have participated in any systematic volleyball training either at the time of the study or in the past (besides apart from regular physical education classes at school, which lasted up to 90 min/week). Body height, weight, and body mass index of the participants are presented in Table 1.

**Table 1.** General characteristics of the participants.

| Variable | EXP Group | | CON Group | |
|---|---|---|---|---|
| | Pre-Training | Post-Training | Pre-Training | Post-Training |
| BH (cm) | 175.2 ± 5.3 | 175.3 ± 4.8 | 174.0 ± 3.9 | 174.4 ± 4.1 |
| BW (kg) | 64.3 ± 5.5 | 64.5 ± 5.2 | 63.2 ± 5.9 | 63.7 ± 5.7 |
| BMI (kg/m$^2$) | 20.3 ± 3.7 | 20.1 ± 2.8 | 20.5 ± 3.2 | 20.8 ± 3.8 |

Values are defined as mean ± SD. Abbreviations: BH, body height; BW, body weight; BMI, body mass index; EXP, experimental; CON, control.

All participants and parents/guardians were familiar with the experimental procedures' possible risks and signed a consent form to participate in the study. The study protocol was carried out in accordance with the Declaration of Helsinki and was approved by the local ethics committee of the University of Novi Sad (Ref. No. 08/2017).

### 2.2. Procedures

Volleyball skills measurements of the subjects in the experimental and control groups were conducted in the school gym. For all subjects, the testing was performed simultaneously in the period from 10 a.m. to 1 p.m. in both the initial and final measurements. All measurements were performed with the same measuring instruments in the initial and final measurements. During the initial and final measurements, the same assistants were

also included. Reliability for the assessment of volleyball skills tests showed to be good, with ICCs from 0.85 to 0.94.

### 2.2.1. Service

The aim of the test is to hit the target on the volleyball court. The player performs ten consecutive serves, trying to direct the ball towards the zone of higher values. Points are awarded according to the specific target areas hit; zero is obtained if the ball hits off the court; also, a higher value is obtained if the ball hits between two zones. The final score is the sum of all ten attempts. Players can choose their desired position behind the service line. The test is a modified version of the test [13], so the test's reliability was performed for this research.

### 2.2.2. Overhand and Forearm Pass

The aim of the test is to hit the target with your overhand and forearm pass from zone VI to position III while the coach is throwing balls from zone VI from the other side of the court. The target is positioned on the net, 3 m from the right sideline. The dimensions of the target are 1.5 m in length and 2 m in width. Players who successfully pass the ball to the target area receive 2 points. The second target area is for balls that did not reach the main target area but would probably reach players in the match situation. The second target area is extended from the right lateral line and is 3 m long and 4 m wide. Players who successfully pass the ball to the second target area get 1 point. Finally, a pass that does not reach the target areas will receive 0 points. The final score is the sum of 6 attempts.

### 2.2.3. Passing

The aim is to hit a horizontal target with your fingers in front of your head [14]. The player must hit a horizontal target in position IV from zone III, with the addition of balls from zone VI, on the same side of the court. The target is placed next to the net at a height of 2.7 m and 5.5 m from the player's position when performing the passing. This target was chosen because it is close to the attacker's position when preparing to spike the ball during the match. The coach is positioned 5 m from the player performing the pass, throwing the ball over his head and passing to the middle player. It is necessary to play the ball with your fingers to a hoop that is 80 cm in diameter. Players who successfully play the ball through the hoop get 3 points. Balls that hit the outside of the hoop and do not pass through the goal are valued 2 points. Players who play the ball 2.3 m from the net (and thus 1.5 m from the goal) get 1 point. Balls that are not in any of the target zones receive 0 points. The final score is the total number of points from 6 attempts.

### 2.2.4. Enjoyment

Students' enjoyment levels in PE were measured using the Sports Enjoyment Scale which is a part of a larger scale, the "Sports Commitment Scale" [15]. We have used only the Sport Enjoyment Scale due to highest reliability and applicability in school settings. The scale included four items and was used to assess the aspects of enjoyment, pleasure, fun, and happiness rated on a 5-point Likert scale that ranged from 1: strongly disagree to 5: strongly agree. The items were modified to represent students' enjoyment in the Volleyball units. The sample items were: (a) "I like volleyball lessons," (b) "I have fun in volleyball lessons," (c) "volleyball lessons make me happy," and (d) "I enjoy volleyball lessons." Scores for the four items were averaged and then used as students' enjoyment scores. The scale has been found to have satisfactory internal consistency in school settings [16].

### 2.2.5. TGfU Model

We have used [17] a model that uses different stages of TGfU and the educational approach of mini-volleyball to allow students to gain an understanding of the game. The mini-volleyball sessions took place twice a week over a period of six weeks for a total of 12 lessons. Each lesson was scheduled for 45 min. The TGfU intervention focused on tactical

problems such as passing to target, using empty space in the attack, serve in different zones. The teacher demonstrates the skills at the beginning of the sessions. Following this teaching procedure, the lesson begins with a modified volleyball game forms (e.g., 1 versus 1 and 2 versus 2 games). The teacher observed the game and investigated tactical problems by stopping the game and asking questions to students, thereby encouraging them to think about the objectives of the game and what their main goals were. Afterward, they returned to the game. Subsequently, the teacher stopped the game and taught game principles based on how the students performed. During lessons, after the demonstration and mini volleyball formats, one game is played. Every lesson has a different game according to the skill that is learned that day. The games used in this study were performed according to Nieves and Luis Estrada Oliver [17]. In the last two weeks, at the end of the sessions, 6 versus 6 games were played on the whole court.

### 2.2.6. Traditional Approach Volleyball Unit

The CON group attended two 45-min traditional volleyball classes per week for 6 weeks. The format of each lesson in a traditional style of teaching was similar. A volleyball class's traditional approach was comprised of several main components: warm-up, instruction or drill practice, and playing 6 versus 6 volleyball game. A typical lesson began with a warm-up that consisted of running and static and dynamic stretches of approximately 5 min, followed by 25 min of instructional drills organized by the teacher, and ending with 15 min of the volleyball game. In the traditional volleyball class, most of the decisions on choice of tasks, team structure, and rate of progression are dictated by the teacher. The teacher delivered the instruction to the whole class as opposed to small group settings.

### *2.3. Statistical Analysis*

Statistical analysis was performed with the SPSS statistical program version 22 (SPSS Inc., Chicago, IL, USA). The results are presented as mean values $\pm$ standard deviation (SD). A was used to demonstrate that the data had a normal distribution ($p > 0.05$). Furthermore, Levene's tests were determined for all test variables. A two-way analysis of variance (ANOVA) was used to test the main effect of the group (EXP vs. CON) and the main effect of time (pre-test vs. post-test), and the interaction of group $\times$ time for volleyball skills test results. The magnitude of the Cohen's d effect (ES) for changes within the group was classified as follows: »trivial« <0.2; »small« 0.2–0.6; »moderate« 0.6–1.2; »large« 1.2–2.0; »very large« >2.0, and »extremely large« >4.0 [18]. A partial eta squared ($\eta^2$) was computed to check the differences between groups, where 0.01 was determined as a small effect, 0.06 as a medium effect, and 0.14 as a large effect [19]. Statistical significance was set at $p \leq 0.05$ level of significance.

### 3. Results

Results from repeated measures ANOVA showed a significant group (EXP vs. CON) $\times$ time (Pre to Post) interaction for overhead passing (F 1, 58 = 5.273, $p$ = 0.025, Partial $\eta^2$ = 0.083) and forearm passing (F 1, 58 = 4.641, $p$ = 0.035, Partial $\eta^2$ = 0.074, See Table 2). When examining the impact of TGfU program on service accuracy, there was a significant main effect for time ($p < 0.01$) with both groups improving their result after the six-weeks intervention (EXP-ES = 0.32, % change = 9.1% vs. CON-ES = 0.57, % change = 14.4%). There were no significant time or group $\times$ time effects for passing ($p > 0.05$).

Perceived sport enjoyment was significantly higher ($p < 0.05$) after the TGfU model (score = 4.15 $\pm$ 0.87 AU) compared to the CON group (score = 3.56 $\pm$ 1.17 AU) (Figure 1).

**Table 2.** Effect of Teaching Games for Understanding (TGFU) on volleyball skills parameters.

| Variable | Group | Pre-Test | Post-Test | ES | % Change | *p*-Value, $\eta^2_p$ |
|---|---|---|---|---|---|---|
| Over-head pass (score) | EXP | 5.1 ± 1.37 | 6.37 ± 1.67 | +0.83 | +24.9% | Group: *p* = 0.633, $\eta^2_p$: 0.004 |
| | CON | 5.77 ± 1.85 | 6.07 ± 1.82 | +0.16 | +5.2% | Time: *p* < 0.001, $\eta^2_p$: 0.193 |
| | | | | | | Interaction: *p* = 0.025, $\eta^2_p$: 0.083 |
| Forearm pass (score) | EXP | 4.33 ± 1.69 | 5.77 ± 2.27 | +1.21 | +33.3% | Group: *p* = 0.077, $\eta^2_p$: 0.053 |
| | CON | 4.2 ± 2.31 | 4.33 ± 1.86 | +0.06 | +3.1% | Time: *p* = 0.012, $\eta^2_p$: 0.104 |
| | | | | | | Interaction: *p* = 0.035, $\eta^2_p$: 0.074 |
| Pass (score) | EXP | 8.77 ± 2.91 | 9.4 ± 3.1 | +0.21 | +7.2% | Group: *p* = 0.143, $\eta^2_p$: 0.037 |
| | CON | 10.03 ± 3.33 | 10.1 ± 2.75 | +0.02 | +0.7% | Time: *p* = 0.403, $\eta^2_p$: 0.012 |
| | | | | | | Interaction: *p* = 0.498, $\eta^2_p$: 0.008 |
| Service (score) | EXP | 25.7 ± 7.96 | 28.03 ± 6.77 | +0.32 | +9.1% | Group: *p* = 0.769, $\eta^2_p$: 0.002 |
| | CON | 25.47 ± 6.59 | 29.13 ± 6.25 | +0.57 | +14.4% | Time: *p* = 0.005, $\eta^2_p$: 0.129 |
| | | | | | | Interaction: *p* = 0.517, $\eta^2_p$: 0.007 |

Abbreviations: TGfU, teaching games for understanding; EXP, experimental group; CON, control group; ES, Cohen d effect size.

**Figure 1.** Enjoyment score in EXP and CON group following 6 weeks. Abbreviations: EXP, experimental group; CON, control group; *, *p* ≤ 0.05 significant differences pre to post-testing.

## 4. Discussion

The present study aimed to implement the TGfU model into the PE program and examine its impact on volleyball skills and enjoyment. The study's main findings were that the six-week PE intervention significantly improved volleyball overhead and forearm passing compared to the CON group. Additionally, the EXP group showed better results for enjoyment compared to the CON group. TGfU is a game-based pedagogical model that generates a greater understanding of the game and increases the engagement, level of PA, motivation, and enjoyment in physical education classes. Studies have mainly used instructional training and a traditional model in order to improve skills in the sport. Moreover, a similar model is used in PE settings in order to teach students the skills of a particular sport. Technical skills like serving, setting, and passing accuracy seem to play a critical role in volleyball performance [20]. Two studies showed similar improvements for small sided volleyball group and instructional training group in volleyball accuracy [14,21]. Gortsila et al. [22], on a sample of young volleyball players after ten weeks of volleyball instructional training, obtained results indicating an improvement in overhead and forearm passing accuracy.

Moreover, Nešič et al. [23] showed certain improvements in the general and specific motor skills (volleyball skills) after three months of applied volleyball practice. Similar results were obtained in PE classes following a precisely defined volleyball program of techniques and tactics [24]. The authors found statistically significant changes in the volleyball technique, and the most significant changes occurred in the tests of serving precision, overhead passing in a circle on the wall, and passing the ball with forearms. Unfortunately, to the authors' knowledge, no study has tried to determine the effects of the TGfU model on volleyball technique in PE settings. However, a recent review showed that TGfU improves several domains like cognitive, motor, and social [25].

Regarding motor development through TGfU model studies González-Víllora [26] and Wang and Wang [27], improved physical and physiological performance was shown, as well as levels of physical activity. Moreover, a novel study [28] showed that a hybrid Sports Education/TGfU volleyball teaching unit permits designing learning environments where students can make decisions and assume responsibilities, perceive themselves as skilled, and establish positive relationships with teammates. One more study [3] confirmed that an instructional model consisting of TGfU, sports education, cooperative learning and peer instruction with a determinant contribution of thoughtful decision-making skills might be key elements to create a powerful learning environment in which students can become self-regulated. The results from the current study showed that the TGfU model improves not only the tactical knowledge [25] but also the technical knowledge, which was confirmed by improved volleyball skills in our students. Therefore, our results and the results of the different studies highlight the TGfU as a positive pedagogical model in PE by promoting learning through understanding and knowledge, thus enjoying the fun of playing, which was confirmed in our study showing better enjoyment than PE classes.

However, one main limitation is that we have used volleyball accuracy tests. Therefore, in future studies, we should use tests with criteria used by volleyball coaches to assess the quality of students' technique and not just outcomes. Moreover, we did not measure the students again to see if retention in learning has occurred. Nevertheless, this is the first study that showed that the TGfU model could improve volleyball skills besides tactical knowledge. Moreover, enjoyment during these classes will motivate students and help them acquire good exercise habits.

## 5. Conclusions

The results show that both teaching models have been effective in improving volleyball skills in closed situations, given that students from both groups reached a similar level of volleyball accuracy. However, given the greater enjoyment favored by the TGfU model, we think it would be preferable to use this model in the PE settings, as a greater level of enjoyment and intrinsic motivation are reached, and these are key factors in the desire to participate in PE and learn skills. Therefore, physical education teachers should consider implementing the mini-volleyball and the TGfU model as an alternative for teaching volleyball in schools.

**Author Contributions:** Conceptualization, N.T. and M.B.; methodology, T.P. and Š.B.; investigation, N.T.; writing—original draft preparation, M.B. and N.T.; writing—review and editing, Š.B., and T.P. All authors have read and agreed to the published version of the manuscript.

**Funding:** This research received no external funding.

**Institutional Review Board Statement:** The study was conducted according to the guidelines of the Declaration of Helsinki, and approved by the the local ethics committee of the University of Novi Sad (Ref. No. 08/2017).

**Informed Consent Statement:** Informed consent was obtained from all subjects involved in the study.

**Data Availability Statement:** Data generated and analyzed during this study are included in this article. Additional data are available from the corresponding author on request.

**Conflicts of Interest:** The authors declare no conflict of interest.

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
