# Peer review of "Effects of Teaching Program Based on Teaching Games for Understanding Model on Volleyball Skills and Enjoyment in Secondary School Students"

_sustainability, doi:10.3390/su13020606_

Round 1
Reviewer 1 Report
Thank you for the opportunity to review this manuscript. The topic is interesting and relevant. I have just two main comments:
- Enjoyment Scale: The scale is cited as the Sports Enjoyment Scale however this scale is one small part of a larger scale the 'Sports Commitment Scale' and is therefore misquoted. Reference needs to be given to the larger scale and why only this part was used in this study.
- A claim is made that the tactical knowledge and technical knoweldge was improved however these were not explicitly tested? Later (Lines241-242) the authors state that 'in future studies we should use tests with criteria used by volleyball coaches to assess students technical skills in school.' Therefore did the authors test for technical skills in this study and what do they mean by this?
Author Response
Reviewer 1
Thank you for the opportunity to review this manuscript. The topic is interesting and relevant. I have just two main comments
Our response: Thank you for your review of our manuscript and for providing some insightful and thought-provoking suggestions to strengthen our manuscript. We feel we have sufficient responses to each of your major concerns listed above, which are further detailed below, and hope that they alleviate the concerns you have regarding the approaches adopted in our manuscript.
- Enjoyment Scale: The scale is cited as the Sports Enjoyment Scale however this scale is one small part of a larger scale the 'Sports Commitment Scale' and is therefore misquoted. Reference needs to be given to the larger scale and why only this part was used in this study.
Our response: We agree with your comment and therefore included the original name of the scale and added the reasons for its usage. In order to better understand sport participants' motivation, the whole scale is recommended. Although Scanlan et al. [https://doi.org/10.1123/jsep.15.1.1], identified six determinants of sport commitment, sport enjoyment was the best predictor of all. Moreover, we wanted to determine only the enjoyment that sports games (volleyball) have in school settings. The common questionnaire used in studies is PACES (https://doi.org/10.1016/j.jshs.2017.09.010). However, this refers to habitual physical activity, and it is more used in adults. The sport enjoyment scale has more applicable in school settings where team games are common during PE classes.
- A claim is made that the tactical knowledge and technical knoweldge was improved however these were not explicitly tested? Later (Lines241-242) the authors state that 'in future studies we should use tests with criteria used by volleyball coaches to assess students technical skills in school.' Therefore did the authors test for technical skills in this study and what do they mean by this?
Our response: Thank you for noticing this mistake. The claim for the tactical knowledge was from other study so we added the reference. We wanted to say that besides tactical knowledge, which is the main goal of TGFU, we were able to improve the tactical knowledge. Additionally, regarding your second issue, we have revised the sentence. The limitation was for the quality of technique and not accuracy. Although they are connected, one is process-oriented and the other is product-oriented skill. Therefore, sorry for this mistake, it was corrected.
Reviewer 2 Report
This paper examines the application of an educational strategy based on TGfU compared to a traditional format for volleyball lessons. The experimental design is appropriate for the research question and the results showed a statistically significant improvement in some aspects of the game for the TGfU group. The paper is well written and referenced appropriately. The paper will be of general interest as TGfU can be applied to other sports. It would be interesting to determine if there is an age effects of the improvements from using TGfU as a teaching methodology. Do all ages respond to the same extent to TGfU? This is not a requirement to publish the paper, but an interesting aspect of the research.
Author Response
Reviewer 2
This paper examines the application of an educational strategy based on TGfU compared to a traditional format for volleyball lessons. The experimental design is appropriate for the research question and the results showed a statistically significant improvement in some aspects of the game for the TGfU group. The paper is well written and referenced appropriately. The paper will be of general interest as TGfU can be applied to other sports. It would be interesting to determine if there is an age effects of the improvements from using TGfU as a teaching methodology. Do all ages respond to the same extent to TGfU? This is not a requirement to publish the paper, but an interesting aspect of the research.
Our response: Thank you for your review of our manuscript and for providing some insightful and thought-provoking suggestions to incorporate in future studies.
Your suggestion is interesting, and we will certainly try to answer this question in the future. Unfortunately, at the moment, we only have the information in this age group.